# Aerial LiDAR Data Augmentation for Direct Point-Cloud Visualisation

**DOI:** 10.3390/s20072089

**Published:** 2020-04-08

**Authors:** Ciril Bohak, Matej Slemenik, Jaka Kordež, Matija Marolt

**Affiliations:** Faculty of Computer and Information Science, University of Ljubljana, Ljubljana 1000, Slovenia; ms4388@student.uni-lj.si (M.S.); jk5456@student.uni-lj.si (J.K.); matija.marolt@fri.uni-lj.si (M.M.)

**Keywords:** LiDAR, point-clouds, point-cloud visualisation, terrain reconstruction, water surface reconstruction

## Abstract

Direct point-cloud visualisation is a common approach for visualising large datasets of aerial terrain LiDAR scans. However, because of the limitations of the acquisition technique, such visualisations often lack the desired visual appeal and quality, mostly because certain types of objects are incomplete or entirely missing (e.g., missing water surfaces, missing building walls and missing parts of the terrain). To improve the quality of direct LiDAR point-cloud rendering, we present a point-cloud processing pipeline that uses data fusion to augment the data with additional points on water surfaces, building walls and terrain through the use of vector maps of water surfaces and building outlines. In the last step of the pipeline, we also add colour information, and calculate point normals for illumination of individual points to make the final visualisation more visually appealing. We evaluate our approach on several parts of the Slovenian LiDAR dataset.

## 1. Introduction

In recent years, aerial data acquisition with LiDAR scanning systems has been used in such diverse scenarios as digital elevation model acquisition [1,2], discovery/reconstruction of archaeological remains [3,4], estimating the vegetation density and/or height [5], etc. While in most scenarios the gathered LiDAR data are used for analysis and digital terrain model development, it can also be used for visualisation. This is especially true for large country-wide LiDAR datasets, which can be augmented with colour information from aerial orthophoto data—https://potree.entwine.io. The number of publicly accessible datasets is increasing; however, they are mostly available for download in the raw (or partly classified) form and are rarely visualised online. Many tools have been developed for point-cloud data visualisation on the web (e.g., Potree [6] and Plasio—https://plas.io or as stand-alone applications (e.g., Cloud Compare—https://www.cloudcompare.org and MeshLab—http://www.meshlab.net), but the LiDAR data are rarely used for direct visualisations due to the many inconsistencies and missing parts which makes them less appealing. LiDAR scans may be incomplete because of:*parts of the acquired objects/terrain are not in the sensor’s line-of-sight and thus cannot be acquired.* For example, vertical building walls, especially in areas with high building density, and mountain overhangs, where parts of the terrain are not visible by the acquisition sensor.*the scanned surface does not reflect, but rather refracts, disperses, dissipates or absorbs light.* For example, water surface, where the laser beam is mostly refracted into and/or absorbed by the water and there is very low to no reflectance back to the sensor.

All of the above-mentioned cases are displayed in Figure 1. Figure 1a shows missing points in a mountain region (as gray background patches within the point-cloud), while Figure 1b shows missing points in building walls, where gray background colour is visible through buildings and on water surfaces (gray background instead of points on the river whining through the city).

In the past, many researchers have addressed the problem of point-cloud reconstruction for specific domains. A method for generating a digital terrain model (DTM) from aerial LiDAR point-cloud data [7] filters out non-ground objects and provides an efficient way of processing large datasets. The approach extends a compact representation of a differential morphological profile vector fields model [8] by extracting the most contrasted connected-components from the dataset and uses them in a multi-criterion filter definition. It also considers areas with the most contrasted connected-components and the standard deviation of contained points’ levels. The output of the method is a DTM defined on a regular grid with high precision. Such a DTM is also used in our approach as an input for estimating the terrain slope. The need for fast terrain acquisition in disaster management led to the development of a LiDAR-based unmanned aerial vehicle (UAV) system [9] equipped with an inertial navigation system, a global navigation satellite system (GNSS) and a low-cost LiDAR system. The data acquired with the presented system were compared with a high-grade terrestrial LiDAR sensor. The results show that the system achieves meter-level accuracy and produces a dense point-cloud representation. While such systems could be used to acquire the missing data in existing datasets, the large amount of hours needed to identify the problematic regions and acquire the missing data are prohibitive. Our approach addresses the problem without the need for additional data acquisition and also provides the identification of the problematic areas for new data acquisitions.

While the above articles address the problem of terrain reconstruction, there are also several works that address the more specific problem of building and urban area reconstruction. In [10], the authors address the problem of a complete residential urban area reconstruction where the density of vegetation is high in comparison to the downtown areas. They present a robust classification algorithm for classifying trees, buildings and ground by adapting an energy minimisation scheme based on 2.5D characteristics of building structures. The output of the system are 3D mesh models. The authors of [11] present a graph-based approach for 3D building model reconstruction from airborne LiDAR data. The approach uses graph-theory to represent the topological building structure, separates the buildings into different parts according to their topological relationship and reconstructs the building model by joining individual models using graph matching. An approach to 3D building reconstruction [12] uses adaptive 2.5D dual contouring. For each cell in a 2D grid overlaid on top of the LiDAR point-cloud data, vertices of the building model are estimated and their number is reduced using quad-tree collapsing procedures. The remaining points are connected according to their grid adjacency and the model is triangulated. An earlier approach [13] produces multi-layer rooftops with complex boundaries and vertical walls connecting roofs to the ground. A graph-cut based method is used to segment out vegetation areas and a novel method—hierarchical Euclidean clustering—is used to extract rooftops and ground terrain.

A more specific problem of roof reconstruction is addressed in [14,15]. Henn et al. present a supervised machine learning approach for identifying the roof type from a point-cloud representation of single and multi-plane roofs, and Chen et al. present a multi-scale grid method for detection and reconstruction of building roofs.

While processing and augmenting the point-cloud data are a hard problem on its own, there is also a growing need for real-time visualisation of large point-cloud datasets on the web. Researchers have developed several solutions for such visualisations that address the problem of multiple visualisation scales, data transfer and others. In [16], authors present a web-based system for visualisation of point-cloud data with progressive encoding, storage and transition. The system was developed for integration into collaborative environments with support for WebGL accelerated visualisation.

A multi-scale workflow for obtaining a more complete description of the captured environment is presented in [17]. The method fuses data from aerial LiDAR data, terrestrial laser scanner data and photogrammetry based reconstruction data in an efficient multi-scale layered spatial representation. While the approach presents an efficient multi-scale layered representation, it does not address the streaming problems that occur in web-based solutions.

Peters and Ledoux [18] present a novel point-cloud visualisation technique—Medial Axis Transform—developed for LiDAR point-clouds. The technique renders the points as circles, whereby it adjusts their radii and orientation. In this way, one can use an order of magnitude fewer points for accurate visualisation of the acquired terrain and buildings. This is very useful in cases where one wants to limit the number of points in visualisation to improve performance.

In recent years, several methods [19,20,21] were developed for real-time progressive rendering of point-cloud data which can also be used for web-based visualisations. The first approach can progressively render as many points as can fit into the GPU memory. The already rendered points in one frame are reprojected and then random points are added to uniformly converge to the final render within a few consecutive frames. The second method supports progressive real-time rendering of large point-cloud datasets without any hierarchical structures. The third method optimises point-cloud rendering using compute shaders. All of the presented methods offer an improvement in terms of performance in comparison to traditional point-cloud rendering.

As real-time direct visualization of large points clouds is already made feasible by the recent progress, in contrast to the presented reconstruction methods, our goal is not to extract 3D mesh models from the point cloud, but to augment the point-cloud data with additional points that would make the visualizations more appealing. To accomplish this, we make use of data fusion of the point cloud data with additional data sources data and present an aerial LiDAR data augmentation pipeline designed to address specific issues of terrestrial point-clouds:*holes on water surfaces*—LiDAR laser beams are mostly refracted into and/or absorbed by the water instead of reflected back to the sensor, which thus creates big holes on surfaces of lakes and rivers,*missing vertical building walls*—in places where due to the direction of the flight and overreach of the roofs the walls do not get scanned and are thus missing in the point-cloud representation, and*holes in mountain overhangs*—in places where mountains are so steep that they form overhangs or where due to the direction of the flight parts of mountains do not get scanned and are thus missing in the point-cloud.

In the rest of the paper, we first present the experimental data in Section 2, the methods for point-cloud data augmentation in Section 3, and results with discussion in Section 4. Finally, in Section 5, we present the conclusions and give pointers for future work.

## 2. Experimental Data

The data used in our experiments have been made publicly accessible by the Ministry of the Environment and Spatial Planning of Slovenia and consist of multiple datasets. In the presented work, we use LiDAR point-cloud data of the Slovenian landscape, orthophoto images of Slovenia, classified vector maps, and the digital terrain model. We give a more in-depth description of individual datasets in the following subsections.

### 2.1. Point-Cloud Data

The point-cloud data from the Slovenian public LiDAR dataset—http://gis.arso.gov.si/evode/ was acquired using the RIEGL LMS-Q780 laser scanner, the IGI Aerocontrol Mark II.E 256 Hz IMU system, and the Novatel OEMV-3 GNSS positioning system at altitudes of 1200 to 1400 m above ground. The postprocessing of the acquired data are presented in depth in the acquisition report [22]. The data were acquired in 2014 and 2015, and were automatically classified into seven classes:unclassified points,ground,low vegetation (up to 1 m),medium vegetation (from 1 to 3 m),high vegetation (above 3 m),buildings,low point (noise).

The classification was performed according to [23], exploiting height distances, the digital terrain model, and geometric properties of neighbouring point sets. The dataset is split into 1 km${}^{2}$ chunks and varies in density depending on the type of landscape (forest, mountains, rural areas, densely populated areas, etc.). On average, the dataset contains five points (first return of LiDAR sensor) per m${}^{2}$. The data are georeferenced in both the Gauß-Krüger coordinate system D48/GK and in the geodetic datum 96 coordinate system D96/TM.

### 2.2. Orthophoto Images

Orthophoto images were obtained from *Portal Prostor*—https://www.e-prostor.gov.si/access-to-geodetic-data/ordering-data/—a repository of publicly available geodetic data. The images are geo-referenced in the same coordinate systems (D48/GK and D96/TM) as the point-cloud data mentioned previously. They are available as small image tiles through a WMTS/REST service, which offers 256 × 256 pixel tiles on 17 levels and the resolution of 0.5 × 0.5 m per pixel on the largest scale.

### 2.3. Classified Vector Map Data

We also make use of the vector outlines of individual buildings from the cadastre dataset and outlines of water surfaces (lakes and rivers). The data (in *shp* [24] file format) were obtained from *Portal Prostor* and are geo-referenced in the same coordinate systems as the point-cloud and orthophoto data.

### 2.4. The Digital Terrain Model

The digital terrain model was also obtained from the Slovenian public LiDAR dataset. The data are stored as a regular grid with a resolution of 1 m, with the corresponding heights.

## 3. Methods

The point-cloud data augmentation pipeline presented in this paper consists of multiple stages. In the first stage, which fuses three data sources, three steps can be performed in parallel: (1) water surface reconstruction, (2) building wall reconstruction, and (3) mountain overhang reconstruction. In the next step, all of the reconstructions are merged into a seamless point-cloud. Finally, we add colour and normal information to make the point-cloud ready for visualisation. The pipeline is presented in Figure 2 and all of the individual steps are presented in the following subsections.

### 3.1. Water Surface Reconstruction

Due to the nature of the acquisition method, the water surfaces usually lack points due to refraction and absorbtion of the LiDAR sensor laser in the medium. This produces holes in larger water surfaces such as lakes, ponds and rivers, as can be seen in Figure 3. For better visualisation, we reconstruct water surfaces by adding points to the point-cloud.

Our algorithm, based on [25], takes the point-cloud data and the 2D polygonal vector representation of water surfaces as input and uses them in the reconstruction process. For every 1 km${}^{2}$ chunk of point-cloud data, we find the water surface polygons that are fully or partly within the chunk and add additional points to the water surface using the Algorithm 1, which is visualised in Figure 4.

For every water surface polygon, we define the height of individual polygon vertices as the average height of its five closest points in the point-cloud. The value of five was selected because it represents a good trade-off between sparse regions (where on average there are 2 points per m${}^{2}$) and dense regions (with 10 or more points per m${}^{2}$).

Next, we generate points within surface water polygons on a regular grid with 1 m spacing, which results in point density of a least 1 point per m${}^{2}$. We remove points that are too close to existing (true) points on the water surface to avoid unwanted point clustering. The added points are randomly shifted along the *x*- and *y*-axes for up to 0.5 m (preserving the desired density), to simulate a more natural distribution of points on the water surface.

To speed up the algorithm, we store the points in a k-d tree structure for fast search of neighbours and distance calculations. The results of the algorithm can be seen in Figure 5.

**Algorithm 1:** Water surface reconstruction algorithm.

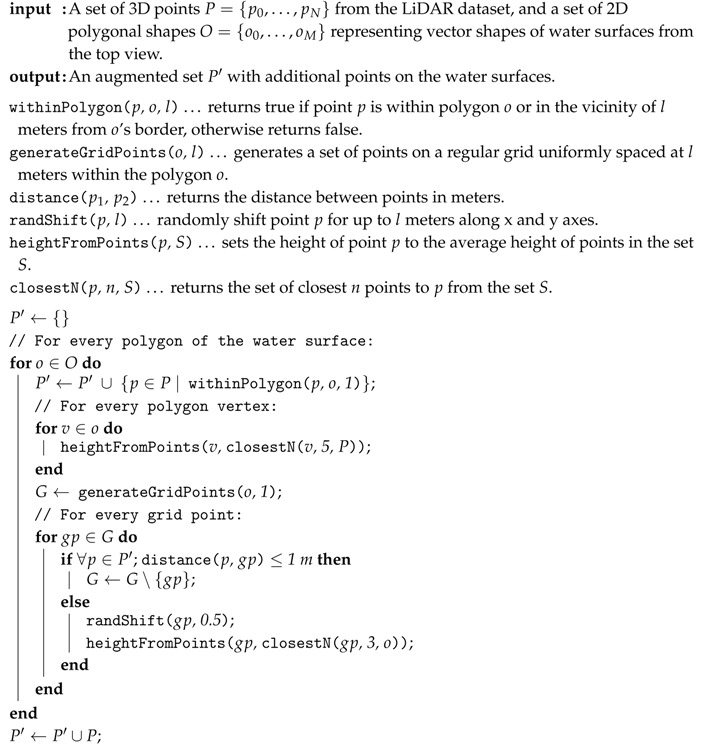



### 3.2. Reconstruction of Building Walls

Points on building walls are usually missing due to their vertical surface which results in shading of the laser beam by either building roofs, neighbouring buildings or due to the steep scanning angle. Examples are shown in Figure 6.

The input of the algorithm consists of point-cloud data, augmented with a 2D polygonal vector representation of building outlines from the cadastre. The building outline polygons in the cadastre are geo-referenced. For every 1 km${}^{2}$ chunk of point-cloud data, we find the building outlines that are partly or fully within the chunk and process them with the Algorithm 2. Individual steps of the algorithm are presented in Figure 7.

For every edge in every building polygon, we generate new equidistant points along the edge according to the average density of the point-cloud chunk—in our case from 2–10 samples/m${}^{2}$.

For preserving the natural look of the data, we randomly shift each generated for up to $\frac{1}{10}$th of the density parameter. The upper value of the random shift was selected arbitrarily, and this step can also be omitted if we want to follow the exact outlines of building polygons.

Next, for every generated point along the edge, we find the closest five point cloud points, taking into account only their *x*- and *y*-coordinates, thus ignoring their height. The value of 5 was selected with the same rationale as in the water surface reconstruction process.

Finally, for all edge points, we generate new points along the *z*-axis starting at the height of the lowest closest point (smallest z-value) and ending at the highest closest point (largest *z*-value). We add all generated points into the point-cloud. The results are visualised in Figure 8.

**Algorithm 2:** The algorithm for reconstruction of building walls.

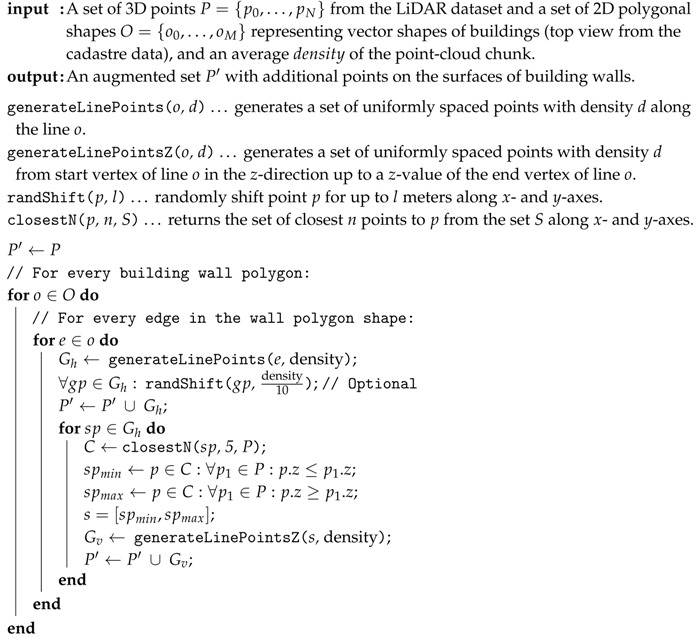



### 3.3. Mountain Overhang Reconstruction

Holes in mountainous terrain occur due to the shading of the laser beam by other parts of the terrain. While many of these errors can be resolved by flying over the same location multiple times and scanning the same part of the terrain from different angles, missing regions still remain. Examples can be seen in Figure 9.

Our approach is based on [26], where the authors presented a method for filling in holes on the surface of 3D point-clouds based on the tangent plane of hole boundary points. The downside of their solution is that the method locates only holes in the point-cloud from the aerial (vertical) perspective; therefore, it misses the holes in the overhanging areas of mountain slopes. To address this, we propose an approach that makes use of the digital terrain model (DTM) data, which was extracted from the LiDAR dataset by Mongus et al. [23]. The DTM was calculated from the same LiDAR dataset and is a 2.5D representation of the surface with heights defined for points on a regular grid (height map). This also means that overhang regions are not visible due to the DTM resolution and because they are *covered* by the ground above. For our purpose, we triangulate the DTM and use the resulting polygon mesh to define the projection planes. We describe our approach below and also illustrate it in Figure 10).

**For each triangle in the DTM mesh representation**:(a)*Project the point-cloud onto the surface tangent plane*, where the tangent plane is defined by the normal of the DTM triangle (see Figure 10a). Due to the 2.5D nature of the DTM, the projection planes cannot exceed the vertical limit.(b)*Find the outer boundary of the projected point-cloud* with the Growth Distance Algorithm [27] to limit the search for holes within the region (see Figure 10b). We also map the points onto a regular grid, with resolution defined by the *density*, to ease the filling process in the following step.(c)*Within the outer boundary find the holes in the point-cloud* using the approach presented in [28], which detects the hole boundaries by using a growth function on the interior boundary of the surface (see Figure 10c).(d)*Add new points within the holes and calculate their 3D positions* using the Catmull–Rom interpolation with boundary points and their neighbors as the basis for interpolation, thus fitting the shape of the filled patches accordingly (see Figure 10d). This process patches regular holes in steep terrain as well as the overhang holes where the hole boundary is extracted well enough. It also patches the holes to some extend in extreme overhang areas, where there is still room for improvement of the algorithm.

The results of the algorithm on examples from Figure 9 can be seen in Figure 11.

### 3.4. Point-Cloud Processing

As our final goal is to directly visualise LiDAR point cloud data, we enhance the scanned point-cloud data with colour and normal information, which can be used to calculate the illumination of points.

We added colour to individual points from the aligned orthophoto data. Each 1 km${}^{2}$ chunk of point-cloud data was aligned with the corresponding orthophoto image of resolution 2000 × 2000 pixels, resulting in resolution 0.5 × 0.5 m per pixel. This was done for all of the LiDAR data chunks in the dataset. To calculate colour, we projected the points in the point-cloud to the xy-plane and took the colour information from the nearest pixel in the orthophoto data. This is also true for points added to building walls and overhangs, even though colours there may not be representative due to the vertical nature of ortophoto imagery.

The results can be seen in Figure 12.

In addition to colour information, we also calculated the normal at each point. Normals are needed to calculate the illumination of points given a light source, which may reflect the time and day of the year, thus yielding appropriate lighting of the rendered scene.

The normals were estimated using the Principal component analysis method as presented in [29], resulting in consistently oriented normals over the entire point-cloud. An example of normal estimation is shown in Figure 13.

### 3.5. Visualisation

To visualise the point cloud, we used an adaptation of the Potree [6] web visualisation framework. Because we are visualising terrain LiDAR data, consisting of points on a relatively flat surface, we changed the octree scene representation into a quadtree representation. This is more meaningful for our type of data, where the amount of vertically overlaid points is negligible in comparison to number of points in the dataset. The adaptation increased loading speeds and simplified data preparation. We also added support for adjusting the light angle according to a selected time and day of the year.

## 4. Results and Discussion

In Figure 14, we present the augmentation results for two sets of data from the Slovenian LiDAR dataset. The first example shows a 1 km${}^{2}$ area of Ljubljana city centre displayed in Figure 14a,c,e, and the second a 6 km${}^{2}$ area of lake Bled with surroundings in Figure 14b,d,f. The first row (Figure 14a,b) displays the input data where points are shaded according to their intensity values. The second row (Figure 14c,d) displays augmentations (water surfaces in green, building walls in blue and terrain reconstruction in red) together with intensity values, and the third row (Figure 14e,f) displays the final output of the proposed augmentation pipeline, where all of the points are merged, the colour information is added from orthophoto images and normal information is calculated using principal component analysis on the point cloud data.

While the results presented above are visually appealing and add a lot of new information in comparison with the original data, in the following section, analyse the results in more detail.

### 4.1. Water Surface Reconstruction Examples

Water reconstruction works well for all surfaces that are present in the vector map dataset. The outlines of those water surfaces are well defined and reconstructions are good especially for lakes and bigger rivers with a uniform river bed. This also holds true for areas under the bridges, since the polygonal representation also covers those parts of terrain as can be seen in Figure 15.

Problems occur with smaller rivers and streams where the water moves the river banks or where the river banks are diverse (e.g., a stream might split into multiple independent streams or smaller islands appear in some parts of the river). These features are not addressed, as they could only be extracted from the orthophoto images for the specific time period when the images were acquired. We also did not take into account intermittent and temporary lakes since they change in size and shape from season to season. This could be addressed with simultaneous acquisition of LiDAR and orthophoto data.

### 4.2. Examples of Reconstruction of Building Walls

Wall reconstruction works well in the majority of cases as can be seen in Figure 16b and other figures displaying the pipeline results. The algorithm may fail in some cases with densely populated buildings because of the incorrectly selected neighbourhood of closest points with minimal and maximal height. This can be observed in Figure 16a.

Other errors may be attributed to inconsistencies between the vector map and the LiDAR datasets. Namely, both are not time-synchronised, which causes missing buildings in the cadastre (new builds) or missing buildings in the LiDAR data (demolished). We could account for some of these errors by considering the automatic classification of LiDAR points into ground, roofs, vegetation, and other classes. However, we ignored this information because of common misclassifications of towers (e.g., church bell towers, castle towers, etc.) for tree canopies and points in building walls into vegetation, as these errors could cause further confusion. Both misclassification types can be seen in Figure 17.

### 4.3. Mountain Overhang Reconstruction Examples

The mountain overhang reconstruction works well for its envisioned use cases (e.g., cliffs, steep terrain, etc.). A very good example is the Bled castle cliff which is mostly missing in the original data but is well reconstructed with the proposed method as can be seen in Figure 18a.

While the primary goal of mountain overhang reconstruction is to add points into holes in the steep terrain, we chose to perform this step on all regions, including the cityscapes. In this way, the proposed algorithm also fills holes in these regions (e.g., missing building walls, smaller water surfaces, etc.). While this was not its intended use, we found that these augmentations were meaningful, so we included them in the final result. Such cases can be seen on the cityscape of Bled town in Figure 18.

## 5. Conclusions

In this paper, we presented a LiDAR point-cloud processing pipeline. Our main goal was to augment parts of the dataset where the acquisition process fails to obtain an appropriate amount of points for direct point-cloud visualisation. We addressed three problematic domains: (1) missing points on water surfaces, (2) missing points on building walls, and (3) missing points in mountain overhang regions. Additionally, we added colour information to the point-cloud from aerial orthophoto images and estimated point normals for calculation of illumination. The proposed pipeline allows for fast and easy augmentation of the point-cloud data, and outputs a denser point-cloud, more suitable for direct visualisation. To the best of our knowledge, this is the first augmentation pipeline that addresses the weaknesses of raw LiDAR point-cloud data for direct visualization.

In our future work, we plan to improve the pipeline in several aspects. We will work on the automatic adaptation of some of the parameters of the proposed algorithms to their context (e.g., wall reconstruction) in order to avoid reconstruction errors. We will develop augmentation algorithms for other problematic features (e.g., bridges, river banks beneath bridges, and roofs). We also plan to preprocess the orthophoto images to remove lighting information and shadows [30], which currently interfere with our illumination calculations.

## Figures and Tables

**Figure 1 sensors-20-02089-f001:**
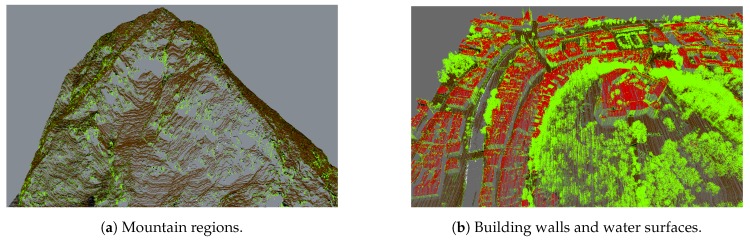
Missing points due to the steepness of the mountain (**a**), in building walls (**b**) and due to refraction and/or absorption on water surfaces (**b**).

**Figure 2 sensors-20-02089-f002:**
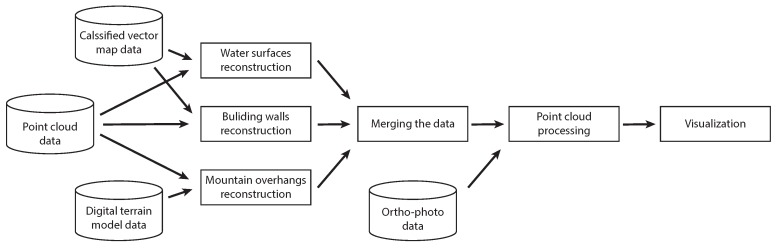
The outline of the point-cloud augmentation pipeline that shows how different input modalities are used for augmenting the point-cloud.

**Figure 3 sensors-20-02089-f003:**
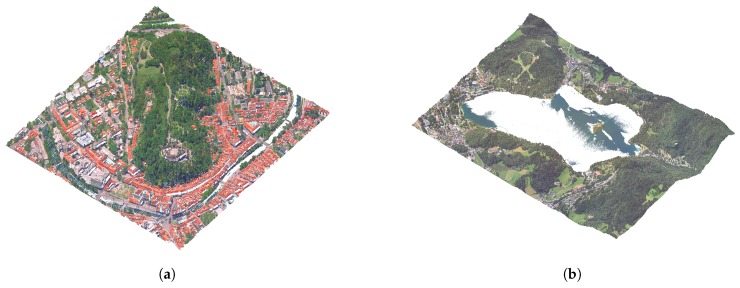
Examples of missing points on water surfaces: (**a**) rivers and (**b**) lakes.

**Figure 4 sensors-20-02089-f004:**
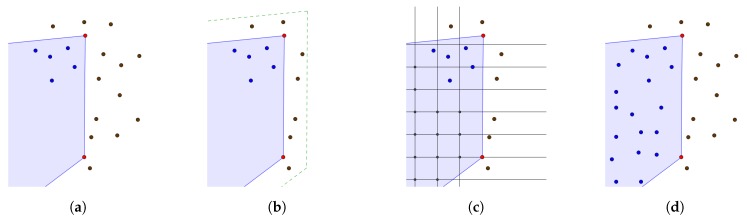
The process of adding points to water surfaces: (**a**) placing polygon data into the point-cloud dataset, (**b**) selecting points in the polygon vicinity, (**c**) adding grid points within the water polygon, and (**d**) removing points that are closer than 1 m to the original points and randomly shifting the added points, where blue points are within the water surface, red points are on the water surface polygon and brown points are terrain points outside the water surface.

**Figure 5 sensors-20-02089-f005:**
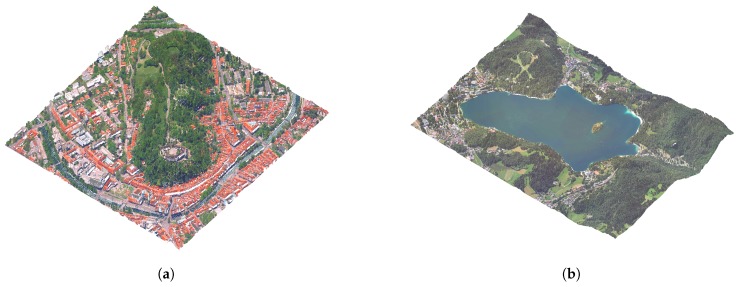
Results of the water surface reconstruction algorithm for examples from Figure 3: reconstructed river in Ljubljana city centre (**a**), and reconstructed Lake Bled surface (**b**).

**Figure 6 sensors-20-02089-f006:**
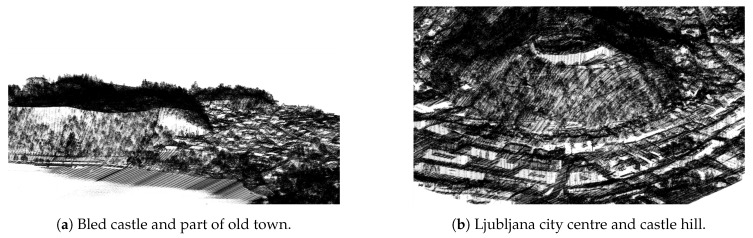
Examples of missing points in building walls, where white regions within the point-cloud indicate the missing points mostly of building walls, but also of terrain surface.

**Figure 7 sensors-20-02089-f007:**
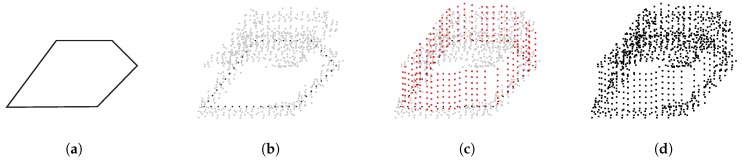
The process of adding points to building walls in the point-cloud using polygon shapes of walls: (**a**) building wall polygons are obtained from the cadastre data and aligned with the LiDAR data, (**b**) the polygon lines are split into segments according to the density parameter, (**c**) for each segment, we find the minimal and maximal height of the closest point cloud points and equidistantly add points in the vertical direction according to the density parameter, and (**d**) we add the generated points to the original point-cloud data.

**Figure 8 sensors-20-02089-f008:**
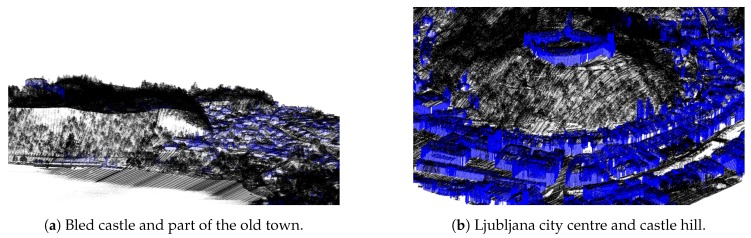
Results of the algorithm for reconstruction of building walls, where blue are the points added by the algorithm and black are the original points from the LiDAR dataset.

**Figure 9 sensors-20-02089-f009:**
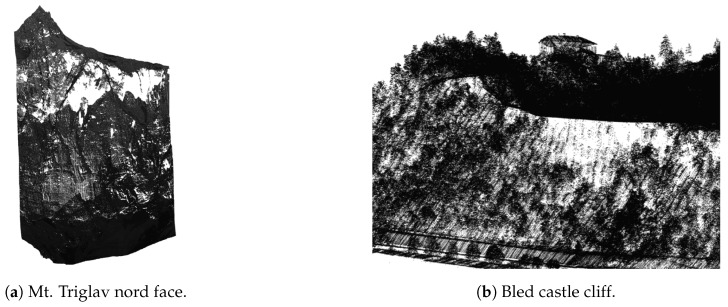
Missing points in mountainous terrain noticable as white areas within the point-cloud.

**Figure 10 sensors-20-02089-f010:**
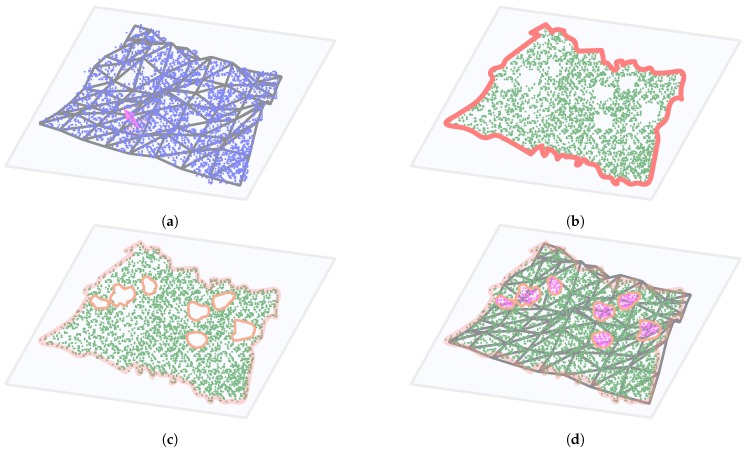
The process of adding points to holes in terrain: (**a**) the points (blue) are projected onto a DTM tangent plane (light blue) defined by the surface normal (pink), (**b**) Outline of the projected points (green) is calculated (red), (**c**) holes in terrain are identified (orange), and (**d**) holes are filled and reprojected back into 3D space.

**Figure 11 sensors-20-02089-f011:**
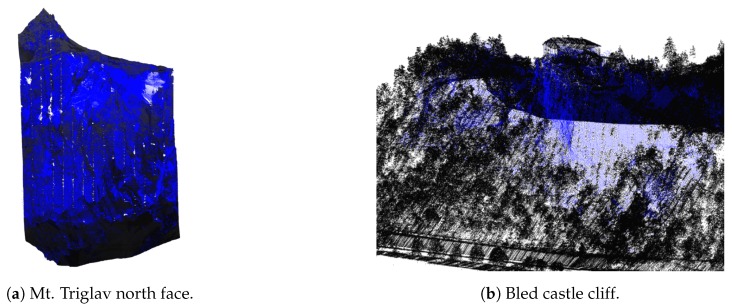
Filling-in missing points in mountain overhangs, where blue are the points added by the presented algorithm and black are the original points from the LiDAR dataset.

**Figure 12 sensors-20-02089-f012:**
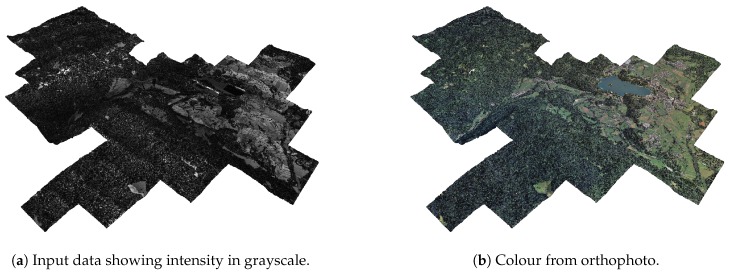
The point-cloud data without colour (**a**) and with colour information (**b**).

**Figure 13 sensors-20-02089-f013:**
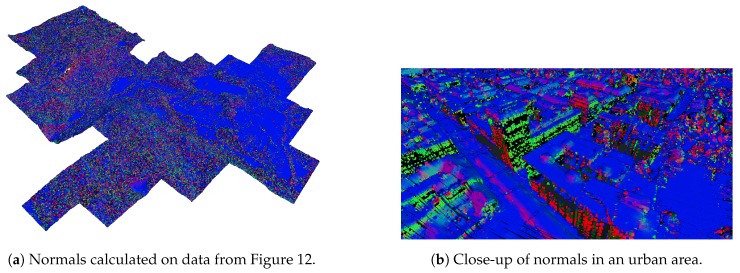
Visualisation of the estimated point normals. x,y, and z components are mapped to red, green, and blue colours, respectively. The colours in the images show the estimated orientation of surfaces on which the points are (e.g., a green point represents a surface whose normal points along the y-axis.).

**Figure 14 sensors-20-02089-f014:**
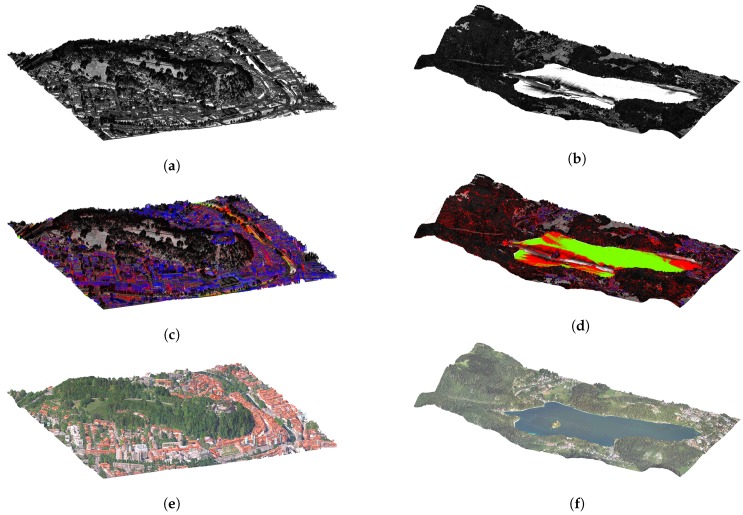
Figure (**a**,**b**) show the input data with intensity values, (**c**,**d**) show the added reconstructions (green—water surfaces, blue—building walls, and red—terrain reconstruction), and (**e**,**f**) show the final results with added colour information from orthophoto images and calculated normals.

**Figure 15 sensors-20-02089-f015:**
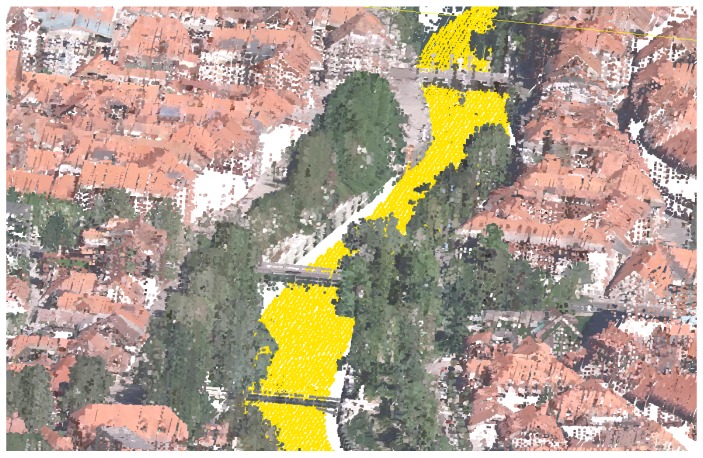
Water reconstruction works well also under the bridges.

**Figure 16 sensors-20-02089-f016:**
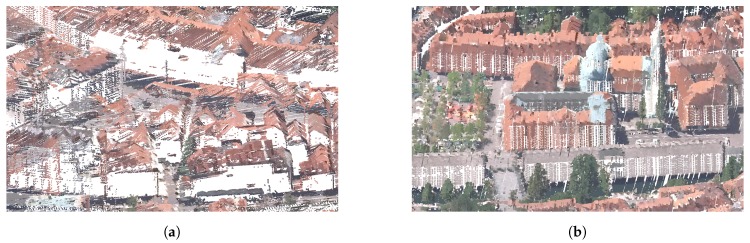
Building walls reconstruction examples for (**a**) bad cases, where the reconstruction fails, and (**b**) good cases, where reconstruction succeeds.

**Figure 17 sensors-20-02089-f017:**
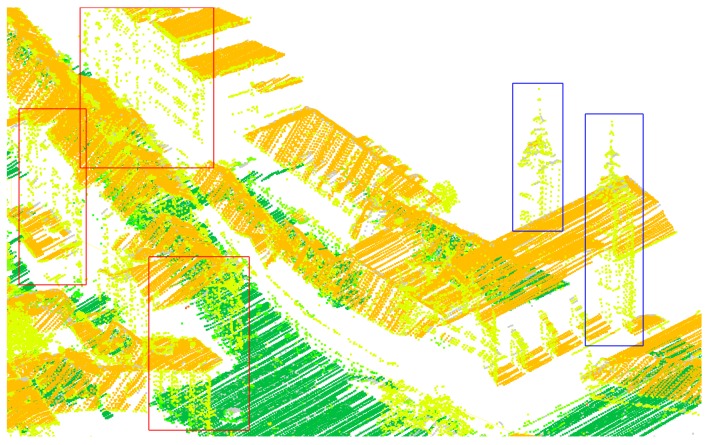
Common misclassification present in the original LiDAR data in Ljubljana city centre where church bell towers are misclassified as trees (blue boxes) instead of building roofs and same is true for points on facades of some buildings (red boxes). Colours of points are: bright green—trees, dark green—ground, and orange—building roofs.

**Figure 18 sensors-20-02089-f018:**
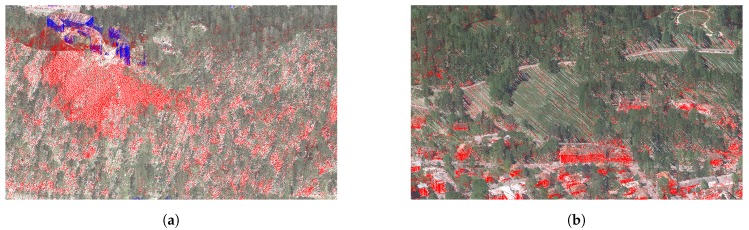
Mountain overhang reconstruction examples: (**a**) points added in the Bled castle hill and (**b**) additional points added on building roofs and walls. Red are points added by the terrain reconstruction, blue are the points added by building walls reconstruction, and other colours are from the orthophoto data.

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
