# Peer review of "Aerial LiDAR Data Augmentation for Direct Point-Cloud Visualisation"

_sensors, 2020, doi:10.3390/s20072089_

Round 1
Reviewer 1 Report
This is a very interesting practical paper addressing visualisation of LiDAR point clouds. It is very well written and very well organised. Therefore easy and pleasant to read.
The methods, with one exception, are described in a repeatable way, with pseudo-code, which is meritorious. The only flaw is the poorer (when not missing) description of the colouring of the building walls and a poor description (although referring to literature) of the filling of holes in the terrain, which seems fuzzy when it comes to solve the problem of the overhangs. Further comments are in the attached PDF.
I congratulate the authors for the present work and hope they can revise the minor flaws indicated.

Reviewer 2 Report
Formal Corrections:
- line 16: a comma is missing before "etc".
- Line 24: I would write the two options in separate paragraphs, together with their examples.
- Fig. 1: Authors should introduce other figures, since what they say is not shown in the figures.
- Line 51: authors cite various papers, ... when in the previous paragraph only one article has been referenced
- Figures 9 and 11 are not very explanatory. They should be more explanatory.
- line 221: 4 steps numbered from 1 to 4 are defined. But in the figure that follows, they are classified as a, b, c, d. Perhaps it would be better to unify the numbering in both sections.
- Figure 18: in the first line of the figure footer, two a) appear ... and there are errors in the description of the colors in each of the images
- The CONCLUSIONS section is a little scarce. Perhaps the utility of everything explained should be further demonstrated. Advantage of the proposed method compared to other methods that attempt to solve this problem should be more evident.
